# Light Isotope Separation through the Compound Membrane of Graphdiyne

**DOI:** 10.3390/membranes12060612

**Published:** 2022-06-13

**Authors:** Valentina A. Poteryaeva, Michael A. Bubenchikov, Alexey M. Bubenchikov

**Affiliations:** 1Department of Mechanics and Mathematics, National Research Tomsk State University, Tomsk 634050, Russia; 2Regional Scientific and Educational Mathematical Center, National Research Tomsk State University, Tomsk 634050, Russia; m.bubenchikov@gtt.gazprom.ru (M.A.B.); bubenchikov_am@mail.ru (A.M.B.); 3Gazprom Transgaz Tomsk, Tomsk 634029, Russia

**Keywords:** light isotope separation, monoatomic membrane, bi-layer membrane, graphdiyne

## Abstract

The separation of isotopes of one substance is possible within the framework of the quantum mechanical model. The tunneling effect allows atoms and molecules to overcome the potential barrier with a nonzero probability. The membranes of two monoatomic layers enhance the differences in the components’ passage through the membrane, thereby providing a high separation degree of mixtures. The probability of overcoming the potential barrier by particles is found from the solving of the Schrödinger integral equation. Hermite polynomials are used to expand all the terms of the Schrödinger integral equation in a series to get a wave function. A two-layer graphdiyne membrane is used to separate the mixture.

## 1. Introduction

Helium and hydrogen have promising applications as new energy sources. Hydrogen is required to operate hydrogen fuel cells and to cool electrical generators. Tritium and helium-3 can carry out thermonuclear fusion. Since the presence of impurities in mixtures is unacceptable, the problem of separating light gases is extremely important. Isotopes of one element differ slightly. Mass-dependent processes such as tunneling are used to separate them. There is a nonzero probability of overcoming the barrier by the components with energy lower than the barrier. It is unattainable for them from the point of view of classical mechanics. The potential barrier should be sufficiently narrow (1–3 nm). Thus monoatomic materials such as porous graphene [1,2,3], graphdiyne [4], carbon nitride [5,6], boron nitride etc. can be used.

The membrane permeability increases sharply when the distance between the two membrane layers is selected equal to the de Broglie wavelength of one of the components (Figure 1). Thus, a resonance effect arises and significantly increases the degree of separation of isotope mixtures [7,8]. The work aims to find conditions (distances between membrane layers and temperature) that provide the highest separation degree of light isotope mixture passing through the compound membrane of two monoatomic layers.

Membranes consisting of two or more layers of monoatomic material have already been synthesized [9,10]. Undoubtedly, maintaining the distance between the layers makes the problem more complicated and requires additional research. However, there are already methods for adjusting the space between layers. For example, changing the interlayer stacking configuration [11], controlling the concentration of electron and hole [12], inducing the structure lattice distortion [13], and the intercalation of active diatomic pairs for establishing the reversible chemical bonding in the interlayer of material [14].

The ability of particles to pass through membranes is described by the wave function. It can be found as the solution of the Schrödinger equation. Usually, the Schrödinger differential equation is solved using numerical methods on a finite interval [15]. The limited interval of variation of the independent variable can affect the accuracy of the results. In this paper, the integral Schrödinger equation is solved, which allows us to obtain a solution on the entire axis. The equation is solved using orthogonal Hermite polynomials. The method based on the expansion of functions into a series of orthogonal polynomials is universal and is used in many problems from various fields [16,17,18,19,20].

## 2. Mathematical Model

The one-dimensional integral Schrödinger equation has the form:(1)$$\psi \left(x\right)-\frac{m}{ik}\int_{-\infty }^{\infty }e^{ik\left|x-x_{0}\right|}U\left(x_{0}\right)\psi \left(x_{0}\right)dx_{0}=e^{ikx}.$$

Here, ψ is the wave function, *m* is the mass of the moving particle, *U*(*x*${}_{0}$) is the potential energy of the membrane, $k=\sqrt{2mE}$, *E* is the energy of the particle. Let us expand the functions participating in (Equation 1) in terms of the Hermite functions [21]:$$\psi \left(x\right)=\sum_{n=0}^{N}C_{n}D_{n}\left(x\right);$$
(2)$$e^{ik\left|x-x_{0}\right|}=\sum_{n=0}^{N}D_{n}\left(x\right)\int_{-\infty }^{\infty }e^{ik\left|\mu -x_{0}\right|}D_{n}\left(\mu \right)d\mu ;$$
$$e^{ikx}=\sum_{n=0}^{N}D_{n}\left(x\right)\int_{-\infty }^{\infty }e^{ik\mu }D_{n}\left(\mu \right)d\mu .$$

Here, *N* is the number of Hermite functions, *D${}_{n}$(x)* is the Hermite function, which is expressed in terms of the Hermite polynomial *He${}_{n}$(x)* as follows:$$D_{n}\left(x\right)=\frac{e^{-x^{2}/4}He_{n}\left(x\right)}{\sqrt{n!\sqrt{2\pi }}}.$$

Substituting (Equation 2) into the integral Equation (Equation 1), we get:(3)$$C_{n}-\frac{m}{ik}\sum_{r=0}^{N}C_{r}\left[\int_{-\infty }^{\infty }\int_{-\infty }^{\infty }U\left(x_{0}\right)e^{ik\left|\mu -x_{0}\right|}D_{n}\left(\mu \right)D_{r}\left(x_{0}\right)d\mu dx_{0}\right]=\int_{-\infty }^{\infty }e^{ik\mu }D_{n}\left(\mu \right)d\mu .$$

Let us introduce the notation:(4)$$G\left(k,n,r\right)=\int_{-\infty }^{\infty }\int_{-\infty }^{\infty }U\left(x_{0}\right)e^{ik\left|\mu -x_{0}\right|}D_{n}\left(\mu \right)D_{r}\left(x_{0}\right)d\mu dx_{0};$$
(5)$$F\left(k,n\right)=\int_{-\infty }^{\infty }e^{ik\mu }D_{n}\left(\mu \right)d\mu =\sqrt{4\pi }i^{n}D_{n}\left(2k\right).$$

Here, G can be considered a square matrix of an order *N*, and F as a column vector. In this case, equality (Equation 3) is a system of linear algebraic equations. We can find coefficients $C_{n}$ in the expansion of the wave function ψ by solving this system. We can rewrite (Equation 3) in matrix notation:(6)$$\left[E_{N}-\frac{m}{ik}G\right]C=F.$$

Some of the integrals in (4,5) can be calculated analytically. Thus, transforming expression (Equation 4), we get:$$G\left(k,n,r\right)=\int_{-\infty }^{\infty }U\left(x_{0}\right)D_{r}\left(x_{0}\right)\left[\sqrt{4\pi }i^{n}D_{n}\left(2k\right)e^{-ikx_{0}}+2iS_{n}\left(x_{0}\right)\right]dx_{0}.$$

Here, it is necessary to calculate only the one-dimensional integral, which takes into account the potential barrier $U(x_{0})$, which has to be solved numerically. However, it can also be calculated with high accuracy since the integration region is limited by the width of the barrier.

The probability density at a point *x* is found as the square of the modulus of the wave function ψ:$$D\left(x\right)={\left|\psi \left(x\right)\right|}^{2}.$$

The value of *D(x)* to the left of the membrane is the probability for particles to pass the membrane.

A large number of atoms and molecules in various states with different energy values are moving through the membrane. Therefore, the Boltzmann distribution for each of the isotopes in the mixture should be taken into account. The Boltzmann distribution gives the probability that a system will be in a certain state as a function of that state’s energy. Thus, the particle distribution function has the following form [22]:(7)$$f\left(E,T\right)=\frac{1}{\sqrt{4\pi kTE}}e^{-E/kT}.$$

Here, *k* is the Boltzmann constant and *T* is the temperature.

In the Boltzmann factor, *E* is the kinetic energy of the separated isotopes. This energy determines the state of the particle; it is uniquely related to its speed. The Boltzmann distribution (Equation 7) can be modeled by a set of particles with velocities (particle energies) varying in a finite range. Based on the results of solving the Schrödinger Equation (Equation 7) with different initial particle energies, we can find the aggregate of values $D\left(x\right)={\left|\psi \right|}^{2}$ for all the values of *x*. However, it is sufficient to know only the value of D(x) behind the barrier since this value determines the transmission coefficient.

For various energies *E*, we obtain the corresponding coefficients for the passage of particles through the membrane, that is, the numerical distribution D(E). Let the index “1” correspond to the component extracted from the mixture. Then the average transmission coefficient of this component can be calculated as follows:(8)$$S_{1}=\int D_{1}\left(E\right)f\left(E,T\right)dE.$$

Similarly, we find a set of solutions to the Schrödinger equation for the retained component and obtain the distribution $D_{2}\left(E\right)$. Therefore, the average value of the transmission coefficient for this component can be determined as follows:(9)$$S_{2}=\int D_{2}\left(E\right)f\left(E,T\right)dE.$$

The degree of separation of a bicomponent isotopes mixture *R* is the ratio of the transmission coefficient of the extracted isotope *S${}_{1}$* to the transmission coefficient of the other component *S${}_{2}$*:(10)$$R=\frac{S_{1}}{S_{2}}.$$

The value of *R* shows how much better one isotope passes through the membrane than another.

## 3. Results

The analytical solution of the Schrödinger equation for some shapes of energy function $U(x_{0})$ is well known through the [23]. The results of the proposed method for the Schrödinger equation solving were compared with the analytical solution as a validation of the mathematical model. Calculations match with an accuracy of 0.001, so a good agreement was obtained [24].

To find the potential energy of the membrane’s impact, we use the integrable version of the Lennard–Jones intermolecular potential [25]:(11)$$\Phi \left(r\right)=4\varepsilon \frac{\sigma }{r}tanh\left[{\left(\frac{\sigma }{r}\right)}^{11}-{\left(\frac{\sigma }{r}\right)}^{5}\right],$$
where $r=\sqrt{{\left(x-x_{0}\right)}^{2}+{\left(y-y_{0}\right)}^{2}+{\left(z-z_{0}\right)}^{2}}$ is the distance between the force centre of the membrane and the moving gas particle, ε is the relative depth of the potential well, σ is the influence radius of the interaction potential. The energy of a monatomic layer of graphdiyne is determined as the result of integration over the infinite surface of the layer. With this potential, the energy values at points close to the layer surface and even on the surface are finite. Thus, in the considered model of wave propagation given by (Equation 1), $U(x_{0})$ is defined as a superposition of the integrals of (Equation 11) over the planes of monatomic layers.

Figure 2a shows potential energy barriers of a bilayer membrane of graphdiyne interacting with helium and hydrogen. The structure of the carbon material is shown in Figure 2b.

The presented problem of separating a mixture of light gas isotopes, namely ${}^{3}$He, ${}^{4}$He, H${}_{2}$, D${}_{2}$, consists in determining the conditions that promote the maximum separation degree of the components. The movement of the mixture of isotopes occurs in the positive direction of the 0x axis perpendicular to the membrane plane.

Calculations were carried out in the temperature range *T* = 30–50 K for all possible values of the distances between the membrane layers *b* from 0 to 5 nm. For each fixed temperature value, the corresponding value of the distance between the membrane layers *b* was found, at which the degree of separation of the mixture *R* turned out to be the maximum (Figure 3).

As can be seen from Figure 3, the calculated maxima of the degree of separation of a two-component mixture are presented in the form of narrow bands. It is typical for resonant separation problems. A potential well appears between two monoatomic graphdiyne layers. The resonant passage of the particles whose de Broglie wavelength is a multiple of the size of the potential well (the distance between the centres of monoatomic layers *b*) occurs. Thus, the selected isotope passes almost freely, while the remaining component may not pass.

Among the obtained results of separation degree maxima for each temperature value R(T), we selected those that provide the maximum degree of separation of the components over the entire temperature range. The results are listed in Table 1.

As can be seen from the table, the degree of separation of a mixture of light gases isotopes is high. The resonance effect helps to achieve this effect. De Broglie wavelengths of isotopes are different because of differences in their mass. When the distance between the peaks of two barriers *b* is a multiple of the de Broglie wavelength of one component and does not coincide with the same parameter of the other isotope, a big difference in the membrane permeability is observed for these components. Choosing the value of *b* that corresponds to the peak permeability of one isotope at the lowest transmission value for the other will provide the best degree of separation (Figure 4b).

A single-layer membrane of a monoatomic material is also capable of isotope mixtures separation (Figure 4a). However, the degree of separation using this membrane is very small. For example, in the case of ^3^He extraction at *T* = 40.1 K, *b* = 3.17 nm, the degree of separation with a single-layer membrane is 8.35. The arising resonance effect in bilayered membrane provides the degree of separation value a few orders of magnitude higher.

Since the properties of the isotopes of one substance, and the isotopes of light gases in general, are close, we focus on the use of resonance transmission for their separation. It is the resonance that explains the high sensitivity of the value of *R* to changes in the distance between the membrane layers and the temperature of the gas environment.

Table 1 allows us to choose the desired temperature and the distance between the membrane layers *b*, which correspond to the maximum passage of a specific component. Thus, the system of two barriers makes it possible to separate isotopes very efficiently.

## 4. Discussion

The results obtained indicate that systems of two barriers corresponding to the bilayer membranes are capable of light isotope separation. An inhomogeneous passage of the mixture’s components through the compound barriers occurs when layers of the membrane are located at a certain distance *b* that corresponds to the de Broglie wavelength of one isotope. Choosing a *b* value that matches the peak permeability of one isotope and the lowest transmission for the other will provide the best result.

The use of Hermite polynomial simplifies to a great extent the problem of solving the Schrödinger integral equation and allows us to find a more accurate solution than numerical methods.

Careful tuning of the parameters will help achieve a separation degree sufficient for industrial needs. The parameters found can be used to construct a bilayered membrane of graphdiyne for light isotope separation.

The described method for determining the membrane permeability applies to the separation of helium or hydrogen isotope mixtures using different monoatomic membranes.

## Figures and Tables

**Figure 1 membranes-12-00612-f001:**
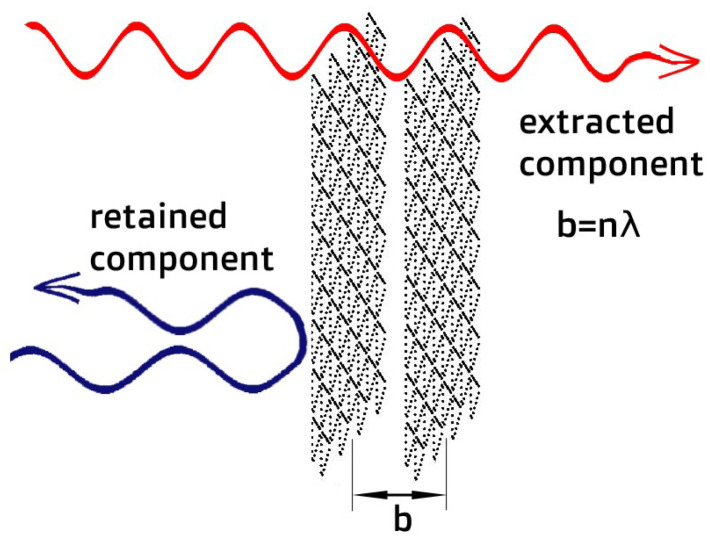
The passing of a mixture of isotopes through the bilayered membrane.

**Figure 2 membranes-12-00612-f002:**
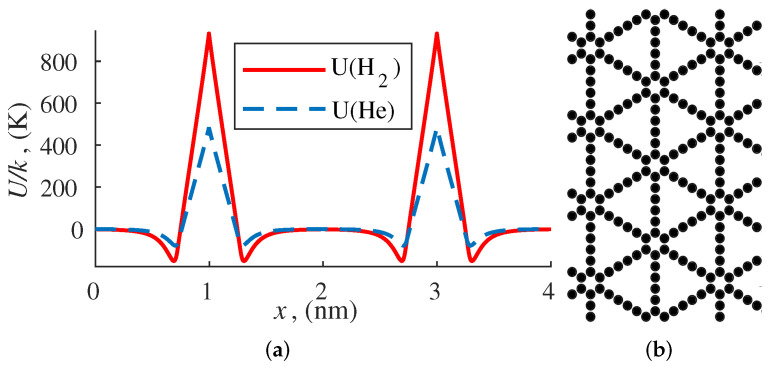
The potential energy of interaction of a bilayer graphdiyne membrane with helium and hydrogen (**a**); the structure of graphdiyne (**b**).

**Figure 3 membranes-12-00612-f003:**
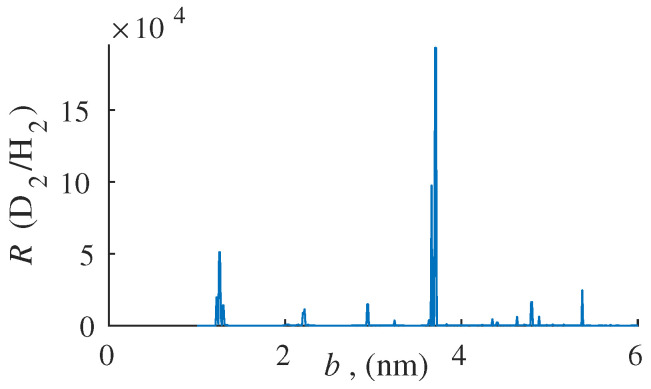
The dependence of the degree of separation *R* on the distance between the membrane layers *b* for hydrogen isotopes, *T* = 45.1 K.

**Figure 4 membranes-12-00612-f004:**
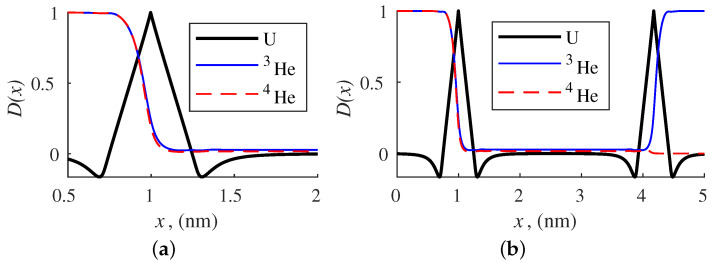
The probability density D(x) for helium isotopes passing through the one-layered (**a**) and two-layered (**b**) membrane of graphdiyne, T=40.1 K, b=3.17 nm.

**Table 1 membranes-12-00612-t001:** The parameters that provide the maximum separation degree of light isotope mixture.

Extracted Component	Retained Component			
	${}^{3}$He	${}^{4}$He	H${}_{2}$	D${}_{2}$
${}^{3}$He	—	*R* = 2.47 × 10${}^{11}$ *T* = 40.1 K *b* = 3.17 nm	*R* = 4.32 × 10${}^{5}$ *T* = 40.1 K *b* = 3.17 nm	*R* = 5.27 × 10${}^{9}$ *T* = 40.1 K *b* = 3.17 nm
${}^{4}$He	*R* = 8.69 × 10${}^{9}$ *T* = 46.5 K *b* = 2.04 nm	—	*R* = 1.53 × 10${}^{6}$ *T* = 46.5 K *b* = 2.04 nm	*R* = 4.54 × 10${}^{9}$ *T* = 46.5 K *b* = 2.04 nm
H${}_{2}$	*R* = 1.17 × 10${}^{11}$ *T* = 37 K *b* = 2.88 nm	*R* = 3.78 × 10${}^{11}$ *T* = 44.9 K *b* = 2.7 nm	—	*R* = 3.73 × 10${}^{12}$ *T* = 44.1 K *b* = 1.8 nm
D${}_{2}$	*R* = 6.75 × 10${}^{8}$ *T* = 30.8 K *b* = 2.08 nm	*R* = 8.37 × 10${}^{10}$ *T* = 30.8 K *b* = 2.08 nm	*R* = 1.9 $\times 10^{5}$ *T* = 45.1 K *b* = 3.7 nm	—

## Data Availability

Not applicable.

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
