# Peer review of "Light Isotope Separation through the Compound Membrane of Graphdiyne"

_membranes, 2022, doi:10.3390/membranes12060612_

Round 1

Reviewer 1 Report

This is an interesting theoretical work presenting separation of isotopes using graphdiyne membrane. The authors used a mathematical model by solving the Schrodinger equation to predict the separation of isotopes. However, the current presentation is confusion to me.

1. I am confused how did the authors use the solved equations to get results in Table 1, which I believe are the main results of this study. What are S1 and S2 to get R? The table only show the "maximum separation degree", but how could readers know it is the maximum? Can the authors provide more evidences?

2. What is the relationship of the modified LJ equation and the mathematical model presented?

3. What is the purpose of Figure 1? Which potential curves are He and H2, respectively? Moreover, a and b is not consistent in the figure and the text.

4. Table 1 is also confusing. Do the authors mean they can either separate 4He from 3He or separate 3He from 4He simply by changing the b value and the T value? That is to say, when b=2.04 nm, and T = 46.5 K, only 4He can pass through the membrane, 3He cannot; in contrast, when b = 1.085 nm and T = 35.3 K, only 3He can pass, 4He will stay? If that is what the authors mean, I would doubt the results unless the authors could provide more evidences.

5. The authors may want to remove the 1st paragraph in Discussion.

Reviewer 2 Report

The one-dimensional Schrödinger equation is solved. However, the model, as well as the problem statement, are not sufficiently explained and substantiated. The values ​​of the separation factors are fantastic and require at least a qualitative verification. In fact, such an effect may well be, but for real application, the model does not correspond to the complexity of the task. In a real system, there will be no one-dimensionality, there will be no Boltzmann distribution, there will be no such potential, with which the authors simulate two layers of the membrane.

-          Not all readers are aware of the notation of quantum mechanics. All designations must be defined in the text, possibly a graphical explanation.

-          The optimal values ​​of the interplanar spacing and temperature should be proved, at least graphically.

-          It is also not clear how the authors are going to make and hold these layers.

-          For what real mixtures the model is expected to be used?

Round 2

Reviewer 1 Report

The current version of the manuscript is much improved. Again, it is an interesting theoretical calculation paper. My last suggestion is to combine Figure 1 and Figure 2, and use a better illustration figure to present the idea. Indeed, Figure 2 is not necessary to be a separate one with no specific purpose or results.

I will suggest the authors to address the following questions in the new figure: (1) how the molecules pass through the graphdiyne layer (pore passage need to be shown); (2) how will be the potential curve affect the molecule transportation.

The current figures are not clear to readers to understand the idea of the authors, so a better illustration will be helpful.

One more suggestion is that, if the authors can compare their results with one layer membrane, it would be more significant.

Author Response

Dear reviewer,

Thank you for your suggestions, they have improved the article.

My last suggestion is to combine Figure 1 and Figure 2, and use a better illustration figure to present the idea. Indeed, Figure 2 is not necessary to be a separate one with no specific purpose or results.

I will suggest the authors to address the following questions in the new figure: (1) how the molecules pass through the graphdiyne layer (pore passage need to be shown); (2) how will be the potential curve affect the molecule transportation.

The current figures are not clear to readers to understand the idea of the authors, so a better illustration will be helpful.

We agree Figure 2 showed a little information. We changed Figure 2 to better illustrate the idea of the work and move it to the introduction as Figure 1. 

We couldn’t merge Figure 1 and Figure 2 because it was not understandable. Figure 1 (Figure 2 now) shows the shape and the height of the potential barriers for different components of the mixture. Figure 2 (Figure 1 now) illustrates the idea of the resonant effect. The isotope passes through the membrane if the distance between the membrane’s layers is multiple to the De Broglie length of this isotope. Another component is retained by the membrane.

We can’t image the pore passage of the molecules because we are solving a one-dimensional problem. We are getting the potential energy U(x) (which describes the interaction of the particles with the membrane) as the integral over the infinite surface of the layer. Thus, we have the averaged energy of the membrane along the 0x axis. 

One more suggestion is that, if the authors can compare their results with one layer membrane, it would be more significant.

Yes, we compared the results of the resonant passage with one-layered membrane in previous works. We added lines 104-108 to address this suggestion and explain Figure 4 in more detail.

Thank you for taking the time to review our paper.

Sincerely, 

Authors.

Reviewer 2 Report

The authors have well responded to my concerns. 

Author Response

Thank you for taking the time to review our paper.

Sincerely, 

Authors.